# Potential Distribution of and Sensitivity Analysis for *Urochloa panicoides* Weed Using Modeling: An Implication of Invasion Risk Analysis for China and Europe

**DOI:** 10.3390/plants11131761

**Published:** 2022-07-01

**Authors:** Tayna Sousa Duque, Ricardo Siqueira da Silva, Josiane Costa Maciel, Daniel Valadão Silva, Bruno Caio Chaves Fernandes, Aurélio Paes Barros Júnior, José Barbosa dos Santos

**Affiliations:** 1Departamento de Agronomia, Universidade Federal dos Vales do Jequitinhonha e Mucuri, Diamantina CEP 39100-000, MG, Brazil; ricardo.ufvjm@gmail.com (R.S.d.S.); josi-agronomia@hotmail.com (J.C.M.); jbarbosa@ufvjm.edu.br (J.B.d.S.); 2Departamento de Agronomia e Ciências Vegetais, Universidade Federal Rural do Semi-Árido, Mossoró CEP 59625-900, RN, Brazil; daniel.valadao@ufersa.edu.br (D.V.S.); brunocaio@ufersa.edu.br (B.C.C.F.); aurelio.barros@ufersa.edu.br (A.P.B.J.)

**Keywords:** biological invasion, climate change, CLIMEX, Ecoclimatic Index

## Abstract

*Urochloa**panicoides* P. Beauv. is considered one of the most harmful weeds in the United States and Australia. It is invasive in Pakistan, Mexico, and Brazil, but its occurrence is hardly reported in China and European countries. Species distribution models enable the measurement of the impact of climate change on plant growth, allowing for risk analysis, effective management, and invasion prevention. The objective of this study was to develop current and future climate models of suitable locations for *U. panicoides* and to determine the most influential climatic parameters. Occurrence data and biological information on *U. panicoides* were collected, and climatic parameters were used to generate the Ecoclimatic Index (EI) and to perform sensitivity analysis. The future projections for 2050, 2080, and 2100 were modeled under the A2 SRES scenario using the Global Climate Model, CSIRO-Mk3.0 (CS). The potential distribution of *U. panicoides* coincided with the data collected, and the reliability of the final model was demonstrated. The generated model identified regions where the occurrence was favorable, despite few records of the species. Sensitivity analysis showed that the most sensitive parameters of the model were related to temperature, humidity, and cold stress. Future projections predict reductions in climate suitability for *U. panicoides* in Brazil, Australia, India, and Africa, and an increase in suitability in Mexico, the United States, European countries, and China. The rise in suitability of China and Europe is attributed to predicted climate change, including reduction in cold stress. From the results obtained, preventive management strategies can be formulated against the spread of *U. panicoides*, avoiding economic and biodiversity losses.

## 1. Introduction

Invasive alien species can reproduce and persist outside the regions to which they are native [1], causing loss of biodiversity [2], economic and social impacts [3], agricultural losses [4], and changes in nutrient stock, soil chemistry, and pH [5,6]. Biological invasion can be influenced by the biological characteristics of the species [7], introduction history [8], anthropogenic factors, and climate change [9].

The level of invasion impact differs across species [10]. African grasses have been introduced as pasture in several countries worldwide, spreading and suppressing native vegetation [11]. Selected traits such as high vigor, seed production, and resistance have become problematic because they also exacerbate invasion [12]. The genus *Urochloa* encompasses species listed among the current primary invasive grasses [11].

*Urochloa panicoides* P. Beauv. is a grass native to Africa that is considered a weed for several crops, such as corn, soybean, and cotton [13]. This species was accidentally introduced in other countries, possibly during cereal cultivation [14]. It is considered one of the most harmful weeds in the United States [15] and Argentina [13], and in Australia, biotypes resistant to atrazine and glyphosate have already been found [16]. Abundant seed production [14] and herbicide resistance facilitate invasion and hinder management in agricultural areas. Invasion by *U. panicoides* has already been reported in Pakistan [17], Mexico [18], and Brazil [19].

The occurrence of *U. panicoides* in China is restricted to the Yunnan province; on the European continent, there are a few occurrences in Belgium and the United Kingdom [20,21]. China is the world’s second largest producer of cotton and corn and is a leading producer of soybean oil; European countries also excel in the world agricultural sector, producing corn and soy oil [22]. Invasion, dissemination, and insertion of species such as *U. panicoides* into agricultural areas would cause losses in biodiversity and productivity and require new weed control strategies, causing an increase in the use of herbicides [23]. The low occurrence of *U. panicoides* in China and Europe is related to seed dormancy when subjected to cold stress [13]. However, climate change, especially changes in temperature, may favor the introduction and establishment of these species [23].

The invasion likelihood of a species depends on its biology, place of origin, and its introduction to the exotic environment [24]. Increases in greenhouse gas emissions, nitrogen deposition, land use changes [25], international commercial transport [26,27], tourism growth [28], socioeconomic development, and rise in gross domestic product of a country [29] are factors that influence the degree of dispersion. In addition, aspects related to physiology, growth [7,24], phenological characteristics [30], and high phenotypic plasticity [7] alter the invasive potential of species.

Among the factors that enhance biological invasion, climate change may alter the competitive potential of an exotic species [31] and growth in places where it currently does not occur [32]. Species distribution models (SDMs) are tools that establish relationships between species occurrence data and predictor variables [33], and they can be used to measure the impact of climate change on the distribution of organisms [34]. Through sensitivity analysis, SDMs determine the main climatic factors that interfere with species growth [35]. CLIMEX software generates SDMs from biological information, occurrence of the target species, and climatic data [36].

Studying the potential and future distribution of possibly aggressive invasive species such as *U. panicoides* allows for risk analyses, effective management, and invasion prevention [32]. The objective of this study was to use the CLIMEX software to predict suitable areas for the establishment of *U. panicoides* based on ecoclimatic conditions, determine potential regions subject to invasion of the species with risk analysis for China and Europe, and establish the most influential climatic parameters for the models.

## 2. Material and Methods

### 2.1. Global Distribution of Urochloa panicoides

Occurrence data for *U. panicoides* were collected in online databases: Global Biodiversity Information Facility [21], based on the record of human observations and occurrences, and in the Invasive Species Compendium [20]. In addition, information on the current distribution of *U. panicoides* was obtained from the published literature, including areas where the species is cultivated or considered invasive (Appendix A). Occurrence records were checked, and those with incomplete or duplicate location information were omitted. A total of 730 occurrences were found and filtered in a 10 km radius, resulting in 355 records.

### 2.2. CLIMEX

The CLIMEX software from climatic and biological parameters predicts a potential distribution of species [36]. Biological parameters are essential components to generate the model, limiting the distribution of the species [37]. The definition of biological parameters was carried out from physiological information of *U. panicoides* and climatic conditions of the places of occurrence.

The growth, stress, and combination parameters define the Ecoclimatic Index (EI), representing areas with climatic suitability for the development and establishment of species. The EI ranges from 0 to 100, where values close to 0 are inappropriate places for the species to grow, and those above 30 are where climatic conditions are considered adequate [38].

### 2.3. Parameter Adjustments and Model Validation in CLIMEX Software

The model for *U. panicoides* was created with parameters related to the biological data of the species and was calibrated as a function of the known distribution. The model validation was based on the distribution of *U. panicoides*, mainly in regions of Australia and South Africa, where higher occurrences were observed. The verification demonstrates reliability in the final model, with most distribution data entered in areas with a high Ecoclimatic Index.

#### 2.3.1. Growth Indices

Thermal requirements of *U. panicoides*, mainly due to germination and dormancy breaking, have already been reported. The low limiting temperature (DV0) used to make the model was 4 °C because the seeds show dormancy [13]. The high limiting temperature (DV3) of 45 °C was defined based on the maximum temperature for germination [13]. The lower optimal temperature (DV1) and upper (DV2) temperatures were established because constant temperatures from 25 °C favor the breaking of dormancy, germination, and growth, and the temperature of 35 °C is considered optimal for the species [13] (Figure 1).

The degree days for *U. panicoides* vary between 1202 and 1723 °C days. Thus, the established value was an average between the two extremes, 1517 °C days [39].

The lower limit of moisture and the ideal soil moisture were determined from the best fit of the model in the global distribution of *U. panicoides* and the upper limit based on moisture content in Queensland, Australia [36]. The lower (SM0), ideal (SM1 and SM2), and upper (SM3) limits established were 0.1, 0.2, 8, and 10, respectively.

#### 2.3.2. Stress Parameters

The cold stress temperature threshold (TTCS) and heat stress temperature threshold (TTHS) were determined from the temperature limits being 4 and 45 °C, respectively [13]. The dry stress threshold (SMDS) was determined according to the lower humidity limit, being 0.01, and the dry stress rate (HDS) was adjusted to −0.01 week^−1^.

Drought and temperature stress parameters were established due to *U. panicoides* being tolerant to low soil moisture; however, seeds were stored dry and at temperatures below 4 °C present dormancy. In addition, a temperature of 45 °C is the maximum temperature for germination [13]. Overall, the parameters were established according to the best fit of the distribution data of *U. panicoides* (Table 1).

### 2.4. Sensitivity Analysis Using CLIMEX

The sensitivity analysis consists of reductions and increases in the adjusted values for the growth and stress indices to determine the parameters with the most significant influence on the model [35]. This analysis makes it possible to establish the variables that cause more substantial interference in the growth and establishment of *U. panicoides* [35].

Sensitivity analysis for *U. panicoides* was performed using CLIMEX software and the 15 model parameters. Temperature-related parameters (DV0, DV1, DV2, DV3, TTCS, and TTHS) had a variation of ±1 °C. Parameters related to moisture, accumulation rates, and drought stress (SM0, SM1, SM2, SM3, TTHS, THHS, SMDS, and HDS) had a variation of ±10%, and degree-days (PPD) had a variation of ±20 °C days [36] (Table 1).

### 2.5. Climate Data, Models, and Scenarios

Modeling in CLIMEX was performed using climate data in a Climond 10’ grid. Monthly minimum average temperature and monthly maximum average temperature, monthly average precipitation, and relative humidity at 09:00 and 15:00 h were used to represent the historical climate (data from 1961 to 1990, centered on 1975) [40].

The same variables were used for future modeling. The global distribution of *U. panicoides* for 2050, 2080, and 2100 was modeled under the A2 SRES scenario using the Global Climate Model (GCM), CSIRO-Mk3.0 (CS) from the Center for Climate Research, Australia [41].

The CS climate system model was chosen because it is comprehensive, encompassing data from the atmosphere, land surface, oceans and sea ice, providing the necessary variables for modeling in CLIMEX (temperature, precipitation and humidity) [41]. The forecasts incorporated into the CS estimate an increase of 2.11 °C and a reduction of 14% in precipitation [42]. Our decision to use A2 SRES was made due to the proven consistency of its premises and incorporation of technological, demographic and economic variables relating to greenhouse gas (GHG) emissions, derived from data representative of the world’s independent, self-reliant countries [42,43,44].

## 3. Results

Distribution data for *U. panicoides* showed 355 occurrence points in 32 countries. Oceania accounts for 77.48% of the reported points, followed by Africa (8.45%), America (7.88%), Asia (5.07%), and Europe (1.12%) (Figure 2a). Approximately 91.83% of the occurrence points were in regions considered highly suitable, of which 6.20% were in moderately suitable regions and 1.97% were in regions unsuitable for the occurrence of *U. panicoides*. The potential distribution of *U. panicoides* coincides with the data collected and agrees with the EI. The model does not predict the climatic suitability of *U. panicoides* in most northern and desert regions, where the species is absent. The results indicate that countries in South America, Central America, Asia, and Europe, with little or no occurrence of the species, have areas with suitable climatic conditions (30 < EI < 100) (Figure 2b).

Approximately 76.62% of these occurrences are concentrated in Australia. Therefore, this region was used to validate the model. The model showed a good fit, with 96.69% of the points in the validation region appearing in areas with highly suitable climatic conditions. This high percentage in the validation area indicated that the model was reliable (Figure 3).

The sensitivity analysis showed that for changes in unsuitable areas, the most sensitive parameters in the model were SM1, SM2, TTCS, THCS, SMDS, and HDS; for low-suitability areas, DV0, DV1, DV2, SM0, SM1, TTCS, THCS, and SMDS; and for high-suitability areas, DV0, DV1, SM0, TTCS, THCS, SMDS, and HDS (Figure 4).

Parameters related to cold stress were found to be sensitive. Sensitivity analyses with lower values of TTCS (3 °C) and THCS (−0.0018 week^−1^) resulted in a reduction of 1.78% and 0.43% in unsuitable areas, an increase of 5.35% and 1.68% in low-suitability areas, and an increase of 2.46% and 0.42% in high-suitability areas, respectively. Sensitivity analyses with higher values of TTCS (5 °C) and THCS (−0.0022 week^−1^) caused an increase of 2.03% and 0.34% in unsuitable areas, a reduction of 6.77% and 1.51% in low-suitability areas, and a reduction of 2.51% and 0.34% in high-suitability areas, respectively (Figure 4).

Changes in temperature and humidity limits also caused changes in model adequacy for the three area classes. DV1 and SM1 values, when reduced to 24 °C and 0.18, caused a reduction of the unsuitable regions by 0.04% and 0.2%; low-suitability areas were reduced by 1.84% and 1.08%; and high suitability areas were increased by 0.87% and 1%, respectively. DV1 and SM1 values, when raised to 26 °C and 0.22, caused an increase in unsuitable areas by 0.03% and 0.2%; low-suitability areas were increased by 2.15% and 0.62%, and high-suitability areas were reduced by 0.98% and 0.78%, respectively. The observed changes in climatic suitability for *U. panicoides* depended on the region under study.

Reductions in areas highly suitable for establishing *U. panicoides* were observed in scenarios designed using CLIMEX under the CSIRO SRES A2 scenario in 2050, 2080, and 2100, compared to those of the current model (Figure 5). The most significant reductions were observed in Brazil, Australia, India, and Africa. The model predicted an increase in low and high suitability areas for establishing *U. panicoides* in Mexico, the United States, Europe, and Asian countries, especially China.

The scenario projected for the year 2100 for China shows an increase mainly in areas highly suitable for establishing *U. panicoides* (Figure 6a,b). The scenario projected in 2100 for Europe foresees the conversion of unsuitable regions to moderately and highly suitable areas for *U. panicoides* (Figure 6e,f). In both areas, compared with the current model, the projected scenario for 2100 showed reductions in cold stress (Figure 6b,d,g,h).

## 4. Discussion

The wide distribution of the grass *Urochloa panicoides* in tropical and subtropical regions is attributed to C4 photosynthetic metabolism [13]. The temperature range in which the growth of *U. panicoides* is maximized (25–35 °C) coincides with the range where plants with the C4 mechanism show maximum CO_2_ assimilation and photosynthesis [45,46].

The validation region (Figure 3) demonstrated the reliability of the model. SDMs generated by CLIMEX are effective tools for determining areas with high climatic suitability for a species [47]. However, limitations occur because the models do not consider biotic interactions such as competition, predation, parasitism, and the occurrence of diseases [48]. Additionally, non-climatic factors such as land use, fertility, and soil type are not considered [49].

The sensitivity analysis in CLIMEX changed the EI, mainly in parameters related to temperature and humidity (TTCS, TTHS, DV0, DV1, SM0, and SM1). Temperature and humidity are the main factors determining growth rate and, consequently, climatic suitability [36]. For *U. panicoides*, constant values below the ideal temperature and lower humidity (25 °C and 0.2) cause seed dormancy, preventing germination and growth [13]. *U. panicoides* has low cold tolerance, especially for prolonged cold periods [50]. Therefore, the parameters of cold stress and accumulation rate are the most sensitive in the analysis. The effectiveness of weed control is influenced by climatic factors [51]. Prolonged high temperatures can change the selectivity and persistence of herbicides [52], making *U. panicoides* difficult to control in regions with high temperatures.

Progressive reductions in the suitability of *U. panicoides* in Brazil, Australia, India, and Africa in scenarios designed using CLIMEX under the CSIRO SRES A2 scenario in 2050, 2080 and 2100 are attributed to possible decreases in humidity and increases in stress during drought [53]. Climate change modifies rainfall patterns, and the water availability influences physiological processes of weeds [51]. Climate change could reduce the suitability of this species in regions such as Australia, where *U. panicoides* herbicide resistance is an economic problem [54].

The increase in suitable areas in Mexico, the United States, and mainly in China and Europe, where there is a low occurrence of the species, occurs because of the reduction in cold stress [53], which is the most sensitive parameter of the model that limits the growth of *U. panicoides*. Temperate regions where the colonization of tropical and subtropical species such as *U. panicoides* is restricted can develop because of the increase in temperature and CO_2_ [55]. Additionally, in the face of climate change, species with C4 metabolism will become more competitive than C3 species [56] and may expand their distribution to higher latitudes [57].

*Urochloa panicoides* is an annual species with considerable herbicide resistance that mainly affects soybean and corn crops [16,58] and is one of the main weeds in cotton cultivation in Australia [54]. Weed populations have specific characteristics; therefore, the spread of *U. panicoides* in China and Europe would change the species composition of agricultural and natural areas [59]. These regions have significant contributions to the world production of cotton, corn, and soybean [22], and the presence of *U. panicoides* would lead to economic and productivity losses in addition to the loss of biodiversity, requiring the creation of integrated management of plant weeds in response to climate change [51,59].

## 5. Conclusions

The SDM for *U. panicoides* generated by the CLIMEX software identified regions, such as Brazil and China, where occurrence is favorable, despite few records.

Sensitivity analysis demonstrated that the most sensitive parameters of the model were related to temperature (DV0 and DV1), humidity (SM0 and SM1), and cold stress (TTCS and TTHS).

The model projections for the years 2050, 2080, and 2100 using CLIMEX under the CSIRO SRES A2 scenario determined reductions in climatic suitability for *U. panicoides* in Brazil, Australia, India, and the African continent and an increase in moderately and highly suitable areas in Mexico, the United States, and European and Asian countries.

The mold projection for 2100 foresees an increase in areas suitable for *U. panicoides* in China and Europe. The introduction and dissemination of this species in these regions can result in biodiversity, productivity, and economic losses.

From the results obtained, it is possible to identify places where *U. panicoides* has high current and/or future climatic suitability and to create management strategies to avoid the entry and spread of the species.

## Figures and Tables

**Figure 1 plants-11-01761-f001:**
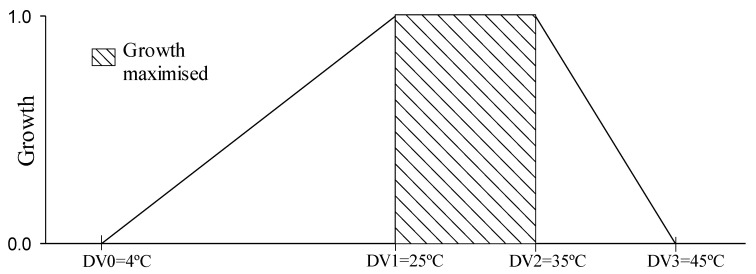
Temperature as a function of population growth. Parameters used to define suitable temperature ranges for *U. panicoides* population growth. DV0: limiting low temperature, DV1: lower optimal temperature, DV2: upper optimal temperature, and DV3: limiting high temperature.

**Figure 2 plants-11-01761-f002:**
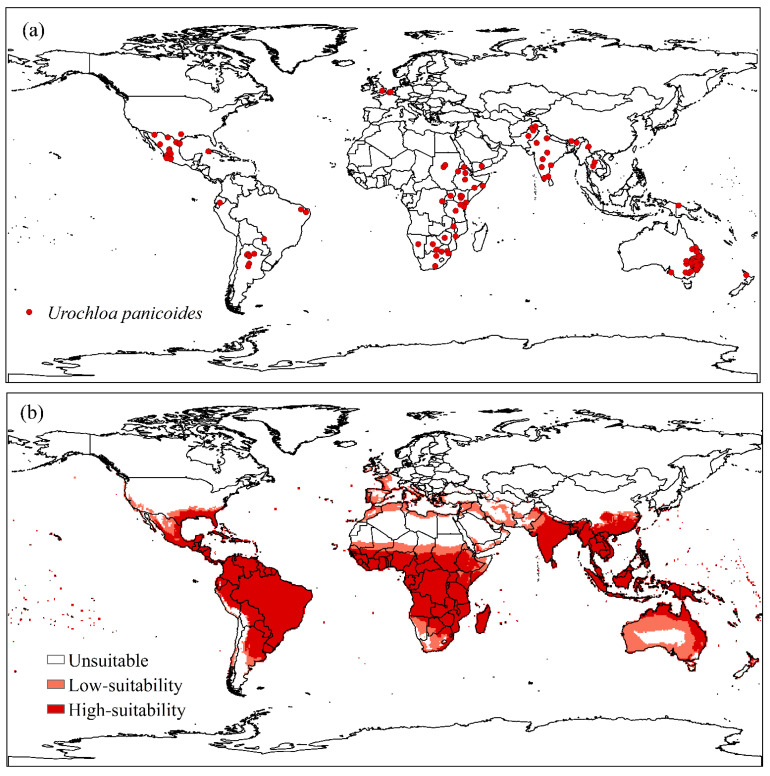
(**a**) Global distribution of *Urochloa panicoides* and (**b**) Ecoclimatic Index (EI) of *U. panicoides*, modeled using CLIMEX. Unsuitable areas in white (EI = 0), low-suitability areas in light red (0 < EI < 30), and high-suitability areas in red (30 < EI < 100).

**Figure 3 plants-11-01761-f003:**
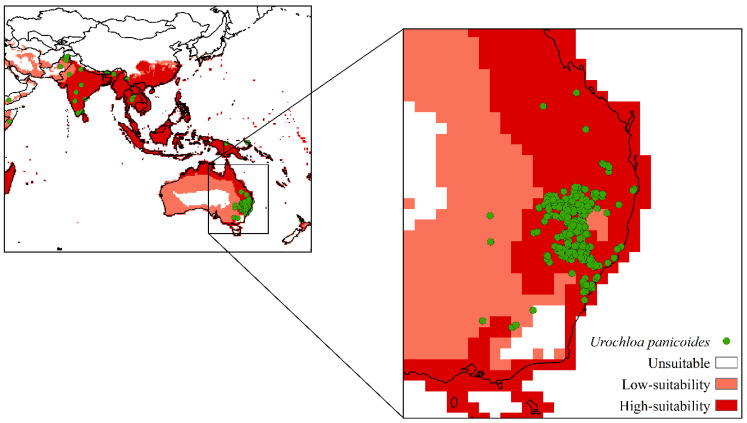
Current distribution of *Urochloa panicoides* in a validation region based on the Ecoclimatic Index (EI). Unsuitable areas in white (EI = 0), low-suitability areas in light red (0 < EI < 30), and high-suitability areas in red (30 < EI < 100).

**Figure 4 plants-11-01761-f004:**
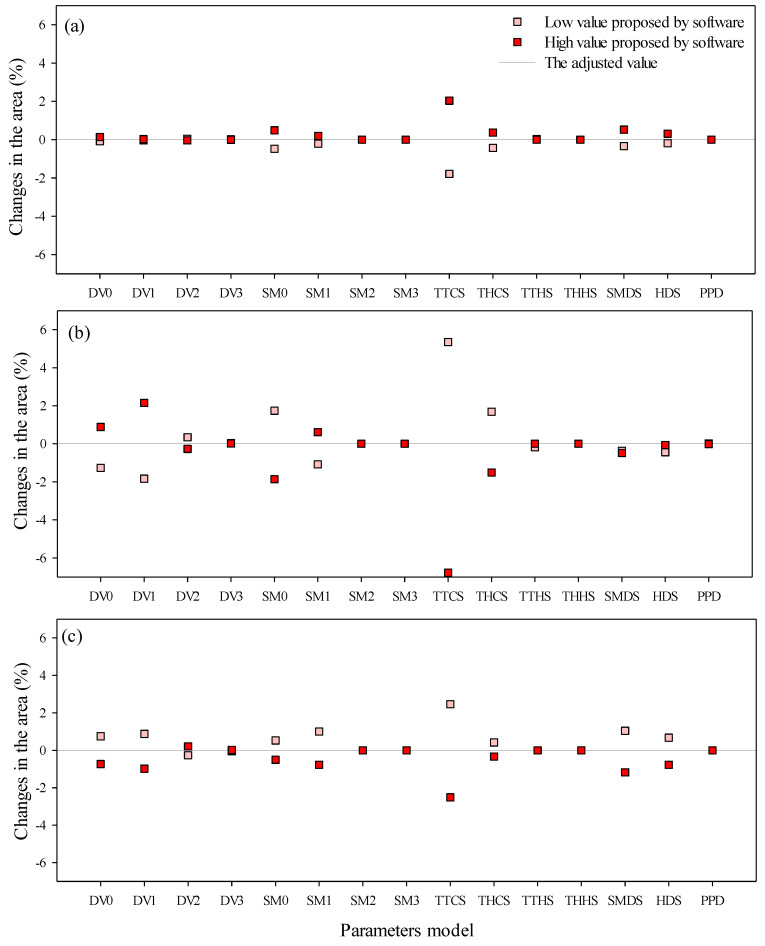
Changes in unsuitable areas (EI = 0) (**a**), low-suitability areas (0 < EI < 30) (**b**), and high-suitability areas (30 < EI < 100) (**c**), in %, for *U. panicoides*, in sensitivity analysis using CLIMEX, based on parameters of greater sensitivity for the Ecoclimatic Index (EI). The values for the parameters used are shown in Table 1.

**Figure 5 plants-11-01761-f005:**
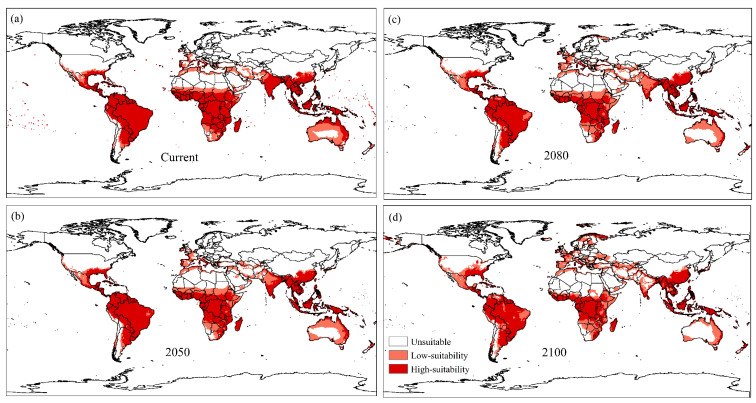
Current (**a**) and projected Ecoclimatic Index (EI) using CLIMEX under CSIRO SRES A2 scenario for the years 2050 (**b**), 2080 (**c**) and, 2100 (**d**), for *U. panicoides*. Unsuitable areas in white (EI = 0), low-suitability areas in light red (0 < EI < 30), and high-suitability areas in red (30 < EI < 100).

**Figure 6 plants-11-01761-f006:**
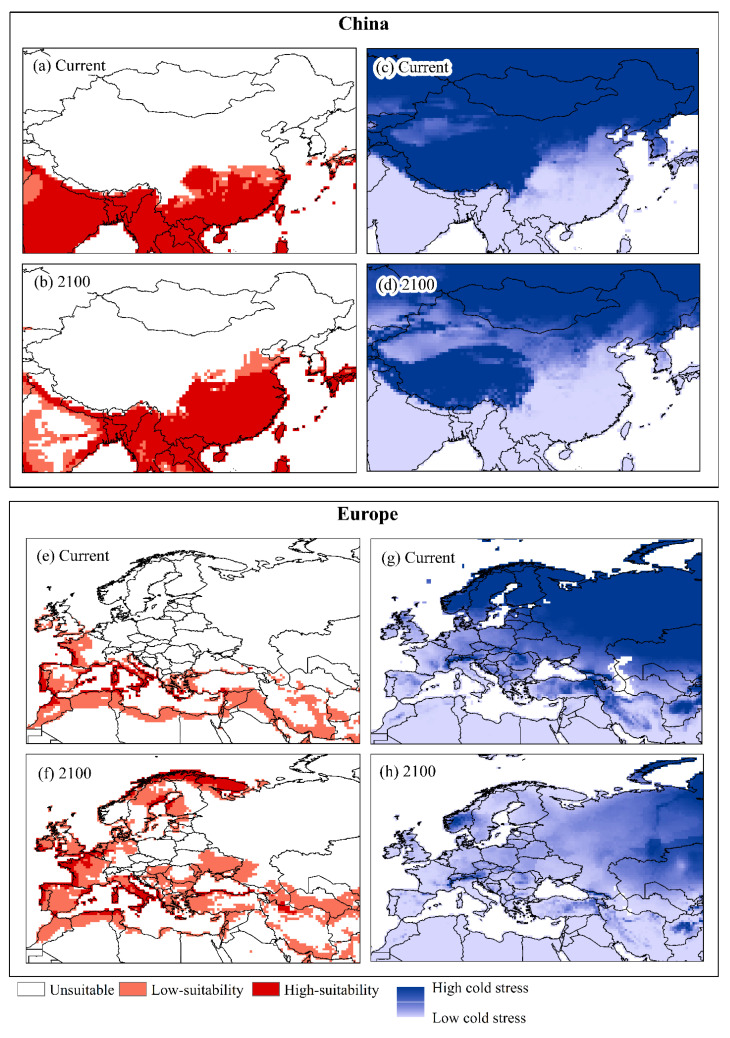
Current and projected Ecoclimatic Index (EI) (**a**,**b**,**e**,**f**) and cold stress patterns (**c**,**d**,**g**,**h**) using CLIMEX under CSIRO SRES A2 scenario for *U. panicoides* for China and Europe. Unsuitable areas in white (EI = 0), low-suitability areas in light red (0 < EI < 30), and high-suitability areas in red (30 < EI < 100).

**Table 1 plants-11-01761-t001:** Adjusted parameter values for modeling *Urochloa panicoides*. Proposed low and high values for the software included in the sensitivity analysis, using CLIMEX.

Parameter	Code	Unit	Low Values	Adjusted Parameter Values	High Values	References
Limiting low temperature	DV0	°C	3	4	5	Ustarroz, 2011; Ustarroz et al., 2015
Lower optimal temperature	DV1	°C	24	25	26	
Upper optimal temperature	DV2	°C	34	35	36	
Limiting high temperature	DV3	°C	44	45	46	
Limiting low moisture	SM0	--	0.09	0.1	0.11	----
Lower optimal moisture	SM1	--	0.18	0.2	0.22	
Upper optimal moisture	SM2	--	7.2	8	8.8	
Limiting high moisture	SM3	--	9	10	11	
Cold stress temperature threshold	TTCS	°C	3	4	5	Ustarroz, 2011; Ustarroz et al., 2015
Cold stress temperature rate	THCS	week^−1^	−0.0018	−0.002	−0.0022	
Heat stress temperature threshold	TTHS	°C	44	45	46	
Heat stress temperature rate	THHS	week^−1^	0.018	0.02	0.022	
Dry stress threshold	SMDS	--	0.09	0.1	0.11	----
Dry stress rate	HDS	week^−1^	−0.009	−0.01	−0.011	
Degree-days	PPD	°C days	1497	1517	1537	Luna, 2018

## Data Availability

All other data are presented in the paper.

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
