# Peer review of "Potential Distribution of and Sensitivity Analysis for Urochloa panicoides Weed Using Modeling: An Implication of Invasion Risk Analysis for China and Europe"

_plants, 2022, doi:10.3390/plants11131761_

Round 1

Reviewer 1 Report

Dear Authors!

I think that manuscript has to be improved and polish prior to its publication. It is kind of standard methodology and procedures, and yet methodological part is not sufficiently explained. It has to be improved. I do not quite understand what do you think when you said that occurrence data were collected in a survey of published literature at the GBIF? There are better sites/databases for survey of the published literature, and there are localities in the GBIF that do not originate from published literature, but still from verified sources. Please explain this initial, and very important, part in more details. In the GBIF there are  some occurrence data from Europe and China older than 2021 when you did your survey, and you state that there Urochloa is not recorded?! With respect to remaining part of the Methods, I feel it is too much a case study of Urochloa based on some species specific knowledge, and it is not straightforward and easy to use it in general. At least this is my impression.

Why some countries in Figure 2 are gray, and some not? Grasslike simbol for occurrences of the Urochloa is not best one for the clarity of the map.

I am not a native English speaker, but it seems to me that some parts of the text could benefit from language editing.

Some specific comments:

There is no need to write "biological invasions" throughout the manuscript. You can mention it once, and latter on just "invasions" will be sufficient.

Line 53-54 - was it introduced intentionally or by accident?

Line 64-65 - what do you mean by "agricultural scenario producing"? "Scenario" here seems to make some confusion.

Line 74-75 - Please rephrase sentence starting with "Species distribution...". It is not quite like that as you have stated.

Line 82 - have you develop a "Climate model" or tested impact of climate on suitable habitats?

Line 95 - cited Jae-Min et al. 2016 is missing in the list of references

References - please check them carefully in terms of their completeness and formatting

Reviewer 2 Report

The manuscript of Duque et al. (Potential distribution an sensitivity analysis for Urochloa..) deals with an interesting aspect of plant conservation and biological invasions, as it is the modeling of the probability of the species distribution, in this case centered in Austalia and China, in order to provide information for managers to implement in the control of the species (U. panicoides).  The manuscript is clear, easy to follow, language is clean, well related results with discussion and figures and tables are appropriate. However, the use of species distribution models is not new, and it has been extensively used in conservation with different software options (Biomapper 4.0, MaxEnt 3.3.3, and a large set of software utilities with basically the same function) but all of them base in the same theoretical concept of probabilities of distribution base in present/absent information of the species and environmental information. In this case the authors are using the CLIMEX with the information provided by CGM, Csiro-mk3 under the A2 scenario. Probably this is one of my main concerns, as long as authors are using the less likely scenario for the predictions. Changes in technology are considered to be poor under this scenario and a dramatic growth of population, with not energy transition. From my point of view and base in the prediction capacity of the last decades of these models, which none of them have been even close of the real data, choosing this scenario is not appropriate. However, from the point of view of the analysis of dispersion of invasive plants, it can be considered of interest.

More specific comments:

-          Lines 72 to 78: Climatic changes are not the only responsible on favoring invasive species colonization of new areas. Some other variables are even more reliable explaining that. Specific elements of globalization that explain the spread of invasive species around the planet are global change (global warming, nitrogen deposition or habitat fragmentation; Dukes and Mooney 1999) but also, and more important, socieconomic ones, such as gross domestic product (GDP; Sharma et al., 2010), transport (Westphal et al., 2008) or tourism (Sutherst, 2000) among others.  Please, avoid speculations, without that, the manuscript is still of interest.

-          Line 92. There are some problems with the figures order. Also, I guess this journal required the Vancouver system of references citation, please check that.

-           Page 5 line 15. I paragraph justifying the selection of the A2 SRES scenario is still necessary.

-          Figure 2, on my version presents very low quality (I am unable tow see if Canary Islands is orange or red). Legend for figure 2a. I am unable to see anything in that figure. I am assuming the figure 2a is the present distribution of U.panicoides, that is far away of the one found in gbif.org. This should be clarified.

-          Figure 5. Much of the presence of U. panicoides at present time in south part of Africa does not appear in Figure 5 a… Is it just the predicted model at present time? If so, the model is not very good predicting that present location (I am again using the information of gbif.org).

-          Discussion. It should be reduced in a 50%... base on your information there is not much discussion of descriptive results (last paragraph and conclusions are ok).

The study is of considerable interest although it is highly descriptive. However, after some minor changes, specially improving the figures, organizing the figures and reduction of discussion length the manuscript can be considered. 

Round 2

Reviewer 1 Report

Dear Authors!

After reading the revised version of the manuscript, I have found just few minor issues to be corrected which I have listed below:

Lines 17-18 - given the added localities in Europe and China, you should change this part of the Abstract

Table 1 - please use decimal point, instead of comma, to be consistent throughout the manuscript

Maps figures - I am still confused why are some countries in gray and some are not. It is not explained in the captions. Please explain, or use same colour for all countries.

Figure 3 - Legend of the lower right map is inversed (mirrored)

Annex 1 - rows 237-244, 284, 286 and 663 are missing name of the localities

With kindest regards!

Author Response

Dear Dr. Reviewer,

Follow the manuscript "plants-1761802" for your appreciation. In this new version, all the suggestions made by the reviewers were made, and the questions were answered. Below are all the points-to-point manuscript changes; they were highlighted in red in the text.

Please do not hesitate to contact me for further information. Thanks in advance for your time and consideration.

Kind regards,

Tayna Sousa Duque*

*Corresponding author: taynaduque24@gmail.com

Reviewers' comments:

Reviewer #1:

Reviewer’s comment: Dear Authors!

After reading the revised version of the manuscript, I have found just few minor issues to be corrected which I have listed below:

Lines 17-18 - given the added localities in Europe and China, you should change this part of the Abstract

Reply: We must express our gratitude to you for the detailed feedback on our manuscript. We thank the reviewer for this important observation. We have revised and amended this information in the text (lines 17-18).

Reviewer’s comment: Table 1 - please use decimal point, instead of comma, to be consistent throughout the manuscript

Reply: Done. These corrections were made in the Table 1.

Reviewer’s comment: Maps figures - I am still confused why are some countries in gray and some are not. It is not explained in the captions. Please explain, or use same colour for all countries.

Reply: We agree and thank you for your observation. There was an error with the file format, some countries were in gray. This has been fixed for all figures in the manuscript.

Reviewer’s comment: Figure 3 - Legend of the lower right map is inversed (mirrored)

Reply: Done. These corrections were made in the Figure 3.

Reviewer’s comment: Annex 1 - rows 237-244, 284, 286 and 663 are missing name of the localities

Reply: Done. We thank the reviewer for this important observation. We have added this information in the Annex 1.
